# Improving Successful Introduction after a Negative Food Challenge Test: How to Achieve the Best Result?

**DOI:** 10.3390/nu12092731

**Published:** 2020-09-07

**Authors:** Joyce Emons, Marije van Gunst, Olivia Liem, Lonneke Landzaat, Nicolette Arends

**Affiliations:** Division Erasmus MC, Department of Pediatrics, Division of Respiratory Medicine and Allergology, University Medical Center Rotterdam, 3015 CN Rotterdam, The Netherlands; m.vangunst@erasmusmc.nl (M.v.G.); o.liem@erasmusmc.nl (O.L.); l.landzaat@erasmusmc.nl (L.L.); n.arends@erasmusmc.nl (N.A.)

**Keywords:** oral food challenge, successful introduction, children, food allergy, allergy, cow’s milk, hens egg, peanut, hazelnut

## Abstract

Oral food challenges (OFC) confirm or exclude the presence of a food allergy. The outcome can be positive (allergic symptoms), inconclusive, or negative (no symptoms). In the case of a negative OFC, parents and children are advised to introduce the challenged food allergen into their diet. However, previous studies showed difficulties in a successful introduction at home. The aim of this prospective non-randomized intervention study is to evaluate the effect of a new strategy with more guidance regarding the dietary introduction after a negative food challenge test. We compared two cohorts: an historical (retrospective) control group of 157 children, previously described, who did not receive any special advice after a negative OFC, versus a new cohort consisting of 104 children, who were guided according to our new strategy of written introduction schemes, food diaries, and several phone calls. In the historical control group, introduction was successful in 56%, partially successful in 16%, and 28% failed to introduce at home. After introduction of our new strategy, complete introduction was found in 82%, 11% had partially introduced, and only 8% failed to introduce the allergen. In conclusion, comprehensive advice and dietary recommendation after a negative OFC results in an increase in successful home introduction. Therefore, more attention, guidance, and follow-up of children and parents are desirable after a negative OFC.

## 1. Introduction

Food allergy is a well-known worldwide health problem. Prevalence numbers vary from 1–11% with patient self-reported food allergy up to 35% [1,2,3]. The most common food allergens in young children are cow’s milk (2.5%), egg (1.3%), peanut (0.8%), wheat (0.4%), soy (0.4%), and tree nuts (0.2%) [4]. The gold standard for the diagnosis of food allergy is an oral food challenge test (OFC). Besides diagnosing an allergy, OFCs are also frequently performed to examine whether tolerance is developed in children who have a history of food allergy. After a positive outcome of a food challenge, a specific diet avoiding the culprit allergen is advised in order to prevent allergic reactions. This diet has a high impact on the quality of life of allergic children and their parents and deserves medical attention and guidance in order to avoid dietary shortage, malnutrition, or excessive avoidance behavior [5,6].

After a negative outcome of a food challenge test, it is recommended that children should (re)introduce the investigated allergen into their diet to improve dietary management and consequently to improve their quality of life. However, usually fewer consultations and follow-ups take place after a negative OFC. Recurrence of allergy is described in patients with a peanut allergy. These patients passed a food challenge test but failed to consume peanut frequently and had a recurrence of their allergy [7,8,9]. In addition, there is increasing evidence that atopic children who avoid allergenic foods for which they are sensitized are at increased risk of developing an acute allergy with the possibility of a severe allergic reaction in such cases [10]. Therefore, unnecessary elimination diets should be avoided as much as possible. For these reasons, a negative OFC can only be considered successful if it is followed by a successful introduction in the diet. Unfortunately, failure of introduction is reported frequently in the literature due to several reasons [9,11,12,13,14]. Reasons for (re)introduction failure are: symptoms during introduction, aversion of the food, fear of the child or parents, habit of not eating the food, other allergies, positive challenge test in patients or parents’ experience, and allergy in the family [11].

The aim of this study is to evaluate whether new comprehensive advice and a written allergen-specific introduction protocol can increase the rate of a successful allergen introduction after a negative OFC for cow’s milk, hen’s egg, peanut, or hazelnut. 

## 2. Materials and Methods 

This prospective non-randomized intervention study was conducted between 16 March and 18 May at the Erasmus Medical Center, Sophia Children’s Hospital in Rotterdam, The Netherlands. Approval of the Dutch medical ethical committee was received (MEC-2016-597). There were 104 children aged 0–18 years with a negative OFC to cow’s milk, hen’s egg, peanut, and hazelnut included in the study. OFCs were either open or double-blinded placebo-controlled. Outcomes were assessed and compared before and after the intervention.

### 2.1. Food Challenges and Intervention

Van der Valk et al. conducted a retrospective study in the same population and clinic from 2008–2013 [11]. A total of 188 negative OFCs were performed in 157 children. None of the children and parents received any special advice after their negative challenge test. The percentage of successful introductions after negative OFCs and reasons of introduction failure were examined. Since this investigated historical group is similar to our enrolled group of children, these results were used as a baseline prior to our intervention. 

For this interventional study, parents and children were asked to participate after a negative OFC with one of the following allergens: cow’s milk, hen’s egg, peanut, or hazelnut (study group). OFC were either open, where the child received an unmasked food (the suspected allergen), or double-blinded (DBPCFC) with the allergen hidden and processed in a matrix. The matrix used for egg, peanut, and hazelnut was gingerbread; for cow’s milk the matrix was soymilk, rice milk, or the hydrolyzed formula the child was using at that time. In the DBPCFC the child received on one day the placebo and the other day the suspected allergen. Blinding was guaranteed for the physician, the nurse, and the patient. Blinding was broken 24 h after the challenge. The food challenge test consisted of a six-step doses regime with increasing dosages every 20–30 min of 1, 3, 10, 30, 100, 300, and 1000 mg protein equivalent. Cumulatively, these dose were comparable to 50 mL of cow’s milk, one fifth of hen’s egg, seven peanuts, or ten hazelnuts. The challenge was discontinued and scored positive when objective allergic symptoms occurred, or subjective allergic symptoms occurred twice on two successive administrations of the challenge material. Objective symptoms and signs were defined as angioedema, urticaria, significant increase in eczema, rash, vomiting, diarrhea, rhinoconjunctivitis, stridor, coughing, wheezing, hoarseness, collapse, tachycardia, and hypotension. Subjective symptoms were defined as exacerbation of generalized itch (in the case of atopic eczema), abdominal pain, nausea and/or cramps, oral allergy symptoms, itchy throat or sensation of throat swelling, difficulty in swallowing, and ‘other’ symptoms such as drowsiness and irritability. Patients were observed for at least 1 h after the last dosage before discharge.

After inclusion, children and parents received a written step-wise introduction protocol concerning the challenged allergen. The protocol contained a list of several products containing the food allergen with stepwise advice on how to introduce carefully and in a well-controlled way. Additionally, parents were asked to fill in a food diary in order to assess the amount and frequency of the introduced allergen (see Appendix A).

The food diary was returned after 6 weeks and evaluated by a telephone consultation with the parents. A questionnaire of 40 questions was carried out during this consultation (see Appendix A). 

### 2.2. Success of Introduction

Level of introduction was categorized into 3 groups: complete introduction, partial introduction, and failed introduction. Complete introduction was defined as regular (at least once a week) unlimited intake of the pure allergen. Partial introduction was defined as consuming small amounts of allergen in pure or processed products. Children with a failed introduction did not succeed in introduction and were still avoiding the tested allergen.

### 2.3. Questionnaires

The questionnaire contained a total of 40 questions (Appendix A). The first part concerned the patient and their family characteristics. The middle part of the questionnaire contained questions regarding symptoms before, during, and after the challenge test. The last part of the questionnaire focused on the successful or failed introduction of the investigated food and the parental experience of the new introduction protocol. 

### 2.4. Data Analysis

Rate of successful dietary introduction was compared between the control group and the study group. Data were collected and processed in IBM SPSS Statistics (Version 25, North Castle, New York, USA). The data were analyzed by means of frequencies, differences, and coherence. Differences in introductions between the control group and the research group were analyzed using a chi-square test. Multivariable logistic regression analysis was performed to study the effect of the intervention and several covariables on the introduction of the allergen. Significance was defined as a *p*-value < 0.05.

## 3. Results

### 3.1. Population

A total of 104 children participated in the current study and 157 children in the control group. No patients were lost to follow-up. Baseline characteristics are presented in Table 1.

In the control group, a total of 188 negative food challenge tests were analyzed for either cow’s milk, chicken’s egg, peanut, or hazelnut. In the study group, a total of 104 food challenge tests were analyzed and performed in the period. Most challenge tests were DBPCFC (73%) and a minority (27%) were open. Patient’s characteristics for both groups are shown in Table 1. The majority of the children (86% and 87%) were sensitized to the tested food allergen (sIgE detectable or positive SPT). Almost half of the patients (41%) had never consumed the allergen before, 41% of the patients had IgE mediated symptoms in their history, 10% had non-IgE mediated symptoms, and 8% did not consume the allergen for a longer period and a sensitization was found. For cow’s milk allergy in the study group, only 45% of patients were sensitized, 27% of patients had symptoms of an IgE mediated cow’s milk allergy, and 73% had non-IgE mediated allergy in their history. In both control and study groups, most children (94% and 90%) had other features of the atopic syndrome (asthma, rhinoconjunctivitis, eczema). In the study group there were more peanut challenge tests and less cow’s milk tests. Patients were on average a little younger and fewer patients had eczema.

### 3.2. Success and Failure of Introduction

A significant improvement of successful introduction was found in the study group compared to the control group (Table 2, *p* < 0.01 and Figure 1). In this study group, failure of introduction was highest in hen’s egg (17%), followed by hazelnut (8%), peanut (3%), and no failure was seen with the introduction of milk after a negative challenge test. In the control group, the highest failure of introduction was seen for peanut (61%), followed by hazelnut (52%), followed by cow’s milk (32%) and the lowest failure of introduction was seen with egg (26%). Reasons for introduction failure are depicted in Table 3.

After introduction of our new strategy, dietary introduction was not successful in only eight children: four patients with hen’s egg, three patients with hazelnut, and one patient with peanut. Egg introduction failed in two cases because of stomach ache and vomiting after eating boiled egg (challenge test was with baked egg), one patient failed due to fear after an anaphylactic reaction to another allergen and the fourth one was unsuccessful because of social issues in the family. Hazelnut introduction failed because of parental interpretation of subjective symptoms during the challenge test that they believed to be caused by the hazelnut (twice) or social issues in the family. The reason for failure of peanut introduction was fear. 

Multivariable logistic regression analysis showed that gender, age, asthma, rhinoconjunctivitis, type of challenge test, and sensitization were not associated with a higher success of introduction. The intervention of our new strategy again was significant in this analysis (*p* = 0.0001). Eczema was found to be associated with successful introduction with an odds ratio of 4.1 (95% CI 1.4, 11.9) (*p* = 0.009), but also the patients with no atopic features were more successful in this analysis (OR 8.1; 95% CI 1.2, 52.2; *p* = 0.00288).

### 3.3. Parental Experience Regarding the New Introduction Protocol

Most parents (74%) reported in the questionnaire that the introduction protocol was clear, informative, and helpful. Furthermore, they reported that this approach contributed to the introduction of the investigated allergen. Two thirds of the parents (65%) reported that the diary was also helpful with introduction. 

## 4. Discussion

This is the first study showing that intensive guidance of allergen introduction after a negative challenge test results in a higher rate of successful introduction at home. In this prospective non-randomized intervention study, a written introduction protocol was used for cow’s milk, hen’s egg, peanut, and hazelnut. Together with the use of a diary and regular phone call appointments, introduction of allergens was significantly improved.

The 82% success rate in this study is higher compared to most other studies in the literature for these allergens. Eigenmann et al. reported 74.6% of successful introduction in 73 patients with a negative food challenge test of several allergens including milk, egg, and peanut [12]. In a Dutch study by van Erp et al., 68% of 103 children with a negative peanut challenge test failed to introduce peanut at home [9]. Whether an introduction is successful also seems to depend on the type of allergen tested, with milk giving the best outcome. Flammarion et al. studied the frequency of recurrent reactions during the introduction of cow’s milk and its consequences for daily life in 67 children and reported a successful introduction in 83% of patients [13]. An additional study that investigated introduction after a negative cow’s milk challenge test reported successful introduction of 80% [14]. In our study milk was best introduced as well, with 100% successful introduction in this group. Perhaps the nutritional importance of daily intake for milk is the explanation for this or less fear for introduction compared to peanut and tree nuts. In addition, in the Dutch diet a lot of dairy products are used like cheese and yoghurt. Comparative studies with high successful rates for milk introduction were performed in France with also a high dairy intake and the Netherlands. Perhaps in Asian and African countries, where the dairy intake is lower, these percentages would be lower. 

All the above studies provide little to no information about the given advice and guidance by the medical staff or dietician after a negative challenge test. Schrijvers et al. studied the effect of personal follow-up and follow-up by phone after a negative cow’s milk challenge test in Dutch children. They found an increase of 22% in both personal and follow-up by phone approaches (91%) compared to follow-up by phone alone (69%) [15]. No additional written advice was given in this study. We hypothesize that the tailored approach for each patient contributed to the success of introduction. The importance of introduction was highlighted for each patient. Patients were able to ask questions at several time points. They were reminded of the introduction in the extra contact moments and in addition in the diary that needed to be completed at home. The food diaries contained examples of food products that helped the parents in the selection of other products in case of food aversion, picky eaters, and dietary habits. The diaries are easy and a good way for the medical staff to check the amount and frequency of introduction with possible symptoms occurring that might influence introduction. 

In addition to the written advice in the protocol and diary, there were also two telephonic consultations in follow-up of the challenge test with the medical staff including a dietician in some of the cases. Additional contact with a dietician resulted in a more successful introduction. An extra telephonic consultation is a good way to check whether introduction has succeeded, to help with problems, and to remind parents to introduce the allergen regularly into the child’s diet.

In both groups the most important cause for a failed introduction were symptoms occurring during introduction at home. This might be caused by a false negative challenge test (due to desensitization during the challenge test) or by symptoms occurring after ingestion of higher dosages or less heated products in the case of milk and egg in which it is known that heating decreases their allergenic potential. Challenge tests for egg were performed with baked (well heated) egg and therefore less heated egg at home could still cause an allergic reaction. Milk OFCs were done with pure non-heated milk. It is important to evaluate these reactions with a pediatric allergist. In a few cases a re-challenge may be necessary for the culprit food. Other causes for a failed introduction were increase in eczema, fear from children or parents, complaints during the food challenge test, or no clear excuse was reported, but parents reported that there was simply no time. The same reasons were reported in other studies as well [9,11]. All the factors described can easily be clarified and addressed if this is acknowledged by the medical staff. In particular, fear is known to be present in a high percentage of allergic patients and this can have a large effect on quality of life. When this is recognized, it can be discussed, introductory steps can be taken more slowly, and psychological help can be offered when necessary.

Patient-related characteristics like asthma, gender, ethnicity, or age did not influence the rate of introduction in this study. This is in contrast to the study of Eigenmann et al. that reported more successful introduction in boys [12]. Another Dutch study regarding cow’s milk introduction after a negative challenge test supported our findings and did not find an association between age and gender on the rate of successful introduction as well [16]. Eczema was found to be associated with more success of introduction. This is surprising since eczema is a chronic disease with frequent exacerbations in time and known by clinicians to complicate the introduction process. Parents confuse eczema with allergy symptoms. It is important for the medical staff to treat eczema properly and aim for optimal control with a dermatologist in consultation when necessary. However, in this study it was not found to be an important risk factor.

The number of patients in this study and in the subgroups consisted of a relatively small number of children, which might have affected the results. However, previously described comparative studies are even smaller. The current study is a prospective non-randomized intervention study and is compared to a previous retrospective study performed in the same hospital. The studies were not blinded and not placebo controlled. Since both groups are from the same hospital with the same medical staff and same food challenge protocols, it is likely that the intervention was the main cause of this increase in successful introduction. 

Finally, our advice is to implement this new strategy in more Dutch Centers where allergic patients are treated and challenge tests are performed to test its national effectiveness. Furthermore, protocols can be translated and adapted to international dietary habits for other countries. In order to keep up to date, it may be possible to realize a digital protocol and/or app to advise patients and parents. 

## 5. Conclusions

Dietary introduction after a negative food challenge is not always successful. Extra comprehensive advice and dietary recommendation from the medical staff results in a significant increase of allergen introduction into the diet. More guidance is advised for follow-up after negative food challenge tests.

## Figures and Tables

**Figure 1 nutrients-12-02731-f001:**
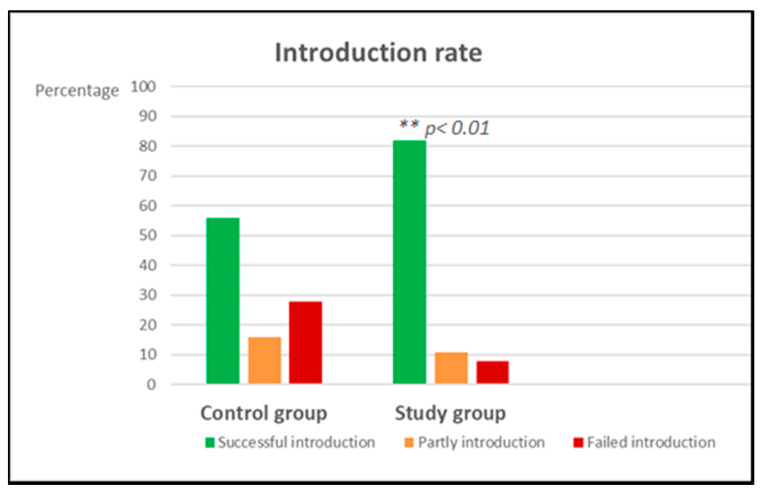
Percentages of success rate of introduction. ** *p* < 0.01.

**Table 1 nutrients-12-02731-t001:** Characteristics of the control and study group.

	Control Group *n* (%)	Study Group *n* (%)	*p* Value
Total number	188	104	
Boys	112 (60%)	69 (66%)	*p* = 0.31
Girls	76 (40%)	34 (33%)	
Age (year)	7.5 (5.5–11.3) *	5.0 (3.0–8.0)	*p* ≤ 0.01
Atopy			
Asthma	63 (40%)	39 (39%)	*p* = 1
Rhinoconjunctivitis	74 (47%)	53 (55%)	*p* = 0.26
Eczema	136 (87%)	74 (74%)	*p* = 0.02
No atopic characteristics	10 (6%)	9 (9%)	*p* = 0.41
Positive Sensitization (SPT/ IgE)	156 (86%)	84 (87%)	*p* = 0.98
Food challenge test			
DBPCFC	146 (78%)	76 (73%)	*p* = 0.46
Open	42 (22%)	28 (27%)	
Tested allergen			
Cow’s milk	41 (22%)	11 (11%)	*p* = 0.04
Egg	39 (21%)	24 (23%)	
Peanut	82 (20%)	32 (31%)	
Hazelnut	70 (37%)	37 (36%)	

* = median (range), SPT = Skin prick test, IgE = Immunoglobulin E, DBPCFC = Double-blind placebo-controlled food challenge.

**Table 2 nutrients-12-02731-t002:** Success of introduction.

	Control Group *N* = 188 (%)	Study Group *N* = 104 (%)	
Successful introduction	106 (56%)	85 (82%)	*p* < 0.01
Partly introduction	30 (16%)	11 (11%)	
Failed introduction	52 (28%)	8 (8%)	

**Table 3 nutrients-12-02731-t003:** Reasons of failed introduction.

	Control Group *N* = 52 (%)	Study Group *N* = 8 (%)	
Symptoms at introduction	12 (23%)	4 (50%)	*p* = 0.15
Aversion of the food	11 (21%)	0	
Symptoms during OFC	2 (4%)	1 (10%)	
Fear for reaction (child)	7 (14%)	2 (20%)	
Dietary habit of avoidance	6 (13%)	0	
Fear for reaction (parents)	5 (10%)	0	
Other/unknown	9 (15%)	1 (10%)	

OFC: Oral food challenges.

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
