# Peer review of "Improving Successful Introduction after a Negative Food Challenge Test: How to Achieve the Best Result?"

_nutrients, 2020, doi:10.3390/nu12092731_

Round 1

Reviewer 1 Report

I have concerns about the OFC protocols used as there is no information in the article about this and methods used, amounts of foods used, how foods were prepared, cooking method etc and how long symptoms were monitored after challenge.

With milk and egg particularly the way it's cooked affects it's allergenic potential; so this is a really big issue and differences in cooking could still cause a valid reaction in a patient - which if cooked/baked has been used in OFC invalidates the study in my opinion.  I refer to milk  ladder/ egg ladder etc as part of evidenced based reintroduction protocols. 

There is no information given about whether participants had had positive RAST IgE or skin prick tests, or type of symptoms experienced and with cows milk allergy if this has been determined to be type 1 or type 2. This information is needed in my opinion to make sense of the results. 

Abstract line 13 two full stops

Line 17  extra space between historical  and control group

Line 38 - sense was a bit hard to determine - would fewer be a better word than less?

Line 66 - can children consent?

Line 71 - what if participants been following milk ladder? 

Line 115 - what are the subjective symptoms?

Line 143 -reported successful re introduction of cows milk -is this down to importance of cows milk/cheese/dairy products in dutch diet - international comparisons useful in discussion to place this in wider context

There wasn't much discussion of findings really at all.

Author Response

We thank the reviewer for the comments. We believe the manuscript is improved after addressing these issues. The revised manuscript is attached.

I have concerns about the OFC protocols used as there is no information in the article about this and methods used, amounts of foods used, how foods were prepared, cooking method etc and how long symptoms were monitored after challenge.

The reviewer is right, there is no information regarding the OFC material and protocols used. We added this information in the paper together with the evaluation of the challenge test and observation time in the methods section on page 2.

With milk and egg particularly the way it's cooked affects it's allergenic potential; so this is a really big issue and differences in cooking could still cause a valid reaction in a patient - which if cooked/baked has been used in OFC invalidates the study in my opinion.  I refer to milk  ladder/ egg ladder etc as part of evidenced based reintroduction protocols. 

We agree with the reviewer that this is important. For milk al challenge test were performed with pure milk. The egg challenge test was performed with gingerbread which includes baked egg. We added this information in the paper and addressed this issue in the discussion section.

There is no information given about whether participants had had positive RAST IgE or skin prick tests, or type of symptoms experienced and with cow’s milk allergy if this has been determined to be type 1 or type 2. This information is needed in my opinion to make sense of the results. 

In table 1 is depicted the number and percentage of people with sensitization to the challenged allergen. The majority of children was sensitized to the tested allergen, 83% in the control group and 77% in the study group. Almost half of the patients (41,3%) never consumed the allergen before, 41,3% of the patients had IgE mediated symptoms in their history, 9.6% non-IgE mediated symptoms and 7.7% did not consume the allergen for a longer period and a sensitization was found. For cow’s milk we investigated our study group for IgE and non-IgE symptoms and found in 27% IgE mediated symptoms and in 73% complaints appropriate for non-IgE mediated cow’s milk allergy. In addition 45% was sensitized to cow’s milk and 55% was not sensitized to cow’s milk. Unfortunately in the control group this was not possible. This information was added to the text.

Abstract line 13 two full stops. One stop is removed.

Line 17  extra space between historical  and control group. Is adjusted.

Line 38 - sense was a bit hard to determine - would fewer be a better word than less? We agree and changed less to fewer.

Line 66 - can children consent? We adjusted this part to parents and adolescents.

Line 71 - what if participants been following milk ladder? 

Patients following the milk ladder were evaluated for their intake of pure milk and baked milk. All milk challenge test were performed with pure milk, the highest step of the ladder. This information is added to the paper.

Line 115 - what are the subjective symptoms?

Subjective symptoms were defined as exacerbation of generalized itch (in case of atopic eczema), abdominal pain, nausea and/or cramp, oral allergy symptoms, itchy throat or sensation of throat swelling, difficulty in swallowing and ‘other’ symptoms such as drowsiness and irritability. We added this information to the paper.

Line 143 -reported successful re introduction of cows milk -is this down to importance of cows milk/cheese/dairy products in dutch diet - international comparisons useful in discussion to place this in wider context.

We thank the reviewer for this interesting point. Indeed in the Dutch Diet we use a lot of dairy products like cheese and milk. We addressed this point in the discussion.

There wasn't much discussion of findings really at all.

We extended the discussion section and hope that the reviewers will find it of interest.

Reviewer 2 Report

In this prospective study Emons et al demonstrated that, after a negative oral food challenge (OFC) test for food allergy, a reintroduction strategy with diaries, pre-compiled schemes and continuous clinical monitoring is more effective than a strategy with no specific suggestion. Main comments:

1) Authors did not discuss about the reason why a tailored approach may be more successful. A biological explanation should be provided.

2) A description of clinical symptoms that led to hypothesize food allergy in children is lacking. Was for example diarrhea reported by the children? Additionally, were symptoms at re-introduction comparable to those before challenge? Please report symptoms details in table 3.

3) A multivariate analysis exploring factors associated with re-introduction failure (for example age, sex, positive SPT/IgE, type of symptom) is lacking.

4) Fear from parents was reduced in the intervention group (0% vs 10%). Therefore it is possible that the careful monitoring in intervention group reassured parents from any fear, reducing the rate of failed re-introduction.

5) Please add p values in tables 1-3.

Author Response

We thank the reviewer for the comments. Addressing this issues resulted in an improvement of our manuscript which can be found attached.

In this prospective study Emons et al demonstrated that, after a negative oral food challenge (OFC) test for food allergy, a reintroduction strategy with diaries, pre-compiled schemes and continuous clinical monitoring is more effective than a strategy with no specific suggestion. Main comments:

1) Authors did not discuss about the reason why a tailored approach may be more successful. A biological explanation should be provided.

The discussion section is expanded and we discussed the success of a tailored approach. We do not exactly understand want the reviewer means with a biological explanation but tried to address this as good a possible.

2) A description of clinical symptoms that led to hypothesize food allergy in children is lacking. Was for example diarrhea reported by the children? Additionally, were symptoms at re-introduction comparable to those before challenge? Please report symptoms details in table 3.

We agree with the reviewer that not much information regarding the history of allergic reactions to the allergen was included. We added this information in the results section page 3. Failure of introduction was only present in 8 patients which are described on page 4. Symptoms during introduction at home were not the same as described in the past.

3) A multivariate analysis exploring factors associated with re-introduction failure (for example age, sex, positive SPT/IgE, type of symptom) is lacking.

A Multivariable logistic regression analysis was performed which showed that gender, age, asthma, rhinoconjunctivitis, type of challenge test and sensitization were not associated with a higher success of introduction. The intervention of our new strategy was again significant in this analysis (p= 0.0001). Surprisingly the presence of eczema in patients was found to be associated with more successful introduction with an odds ratio of 4.1 [95% CI 1.4, 11.9] (p= 0,009). In addition, patients with no atopic features were more successful in this analysis as well (OR 8.1; 95% CI 1.2,52.2; p= 0,00288). Eczema was more present in the control group with a lower success rate of introduction so this cannot explain the strong effect of our new intervention strategy. We added this analysis to the result sections and described it in the discussion section.

4) Fear from parents was reduced in the intervention group (0% vs 10%). Therefore it is possible that the careful monitoring in intervention group reassured parents from any fear, reducing the rate of failed re-introduction.

We agree with the reviewer that fear was reported as an imported factor for failure. We addressed this point extra in the discussion section page x, line x-x

5) Please add p values in tables 1-3.p values were added to all tables like suggested.

Reviewer 3 Report

The authors have taken on an important practical area of allergy diagnosis and prevention.  If someone is suspected of food allergy to some important allergenic source, and then by food challenge (open or DBPCFC) they are shown to be non-reactive (hopefully to a high enough dose, they should regularly consume the food to ensure that they remain allergy free.  At least that is the dogma of the peanut work in the last 10 years, and in some other work.  It actually follows logically from human experience. Tolerance is an acquired immune response.

The down-side as noted by others in this paper, is that often the medical unit does not provide enough instruction and especially not enough follow-up to the family to ensure that the food is added to the diet at some regularity.

One lack of clarity is a lack of description of dose of the food challenge.  Not that it requires a long, detailed discussion, but there should be reference to adequate dose and timing of food challenge.  Sufficient to prove lack of reactions, in a safe manner.

A second missing is the dose and repeat of continued consumption. There are some publications that address this.

An important missing is how many people of the control study, and then the actual study were + to the challenge material? 

A final point that should be added is an English translation of the questionaire, that is provided as a WORD file.  It is all in Dutch.  I do not read Dutch, and I think that most MDPI readers do not either.  So I recommend an English translation as well as the original Dutch.

Author Response

We thank the reviewer for the comments. Addressing this issues resulted in an improvement of our manuscript which can be found attached.

The authors have taken on an important practical area of allergy diagnosis and prevention.  If someone is suspected of food allergy to some important allergenic source, and then by food challenge (open or DBPCFC) they are shown to be non-reactive (hopefully to a high enough dose, they should regularly consume the food to ensure that they remain allergy free.  At least that is the dogma of the peanut work in the last 10 years, and in some other work.  It actually follows logically from human experience. Tolerance is an acquired immune response.

The down-side as noted by others in this paper, is that often the medical unit does not provide enough instruction and especially not enough follow-up to the family to ensure that the food is added to the diet at some regularity.

One lack of clarity is a lack of description of dose of the food challenge.  Not that it requires a long, detailed discussion, but there should be reference to adequate dose and timing of food challenge.  Sufficient to prove lack of reactions, in a safe manner.

The reviewer is right, there is no information regarding the OFC material and protocols used. We added this information in the paper together with the evaluation of the challenge test and observation time, method section page 2.

A second missing is the dose and repeat of continued consumption. There are some publications that address this.

Level of introduction was categorized into 3 groups; complete introduction, partial introduction and failed introduction. Complete introduction was defined as regularly (at least weekly) unlimited intake of the allergen. Partial introduction was defined as consuming small amounts of allergen in pure or processed products. Children with a failed introduction did not succeed in introduction and were still avoiding the tested allergen

An important missing is how many people of the control study, and then the actual study were + to the challenge material? 

We included 104 children in this current study with a negative OFC and compared them to an historical cohort with 157 children undergoing 188 challenge tests. We added this number to the material and methods section, together with the tables. There were no dropouts or loss to follow up in the study.

A final point that should be added is an English translation of the questionaire, that is provided as a WORD file.  It is all in Dutch.  I do not read Dutch, and I think that most MDPI readers do not either.  So I recommend an English translation as well as the original Dutch.

We agree with the reviewer and an English translation is added to the new submission as supplementary data. In addition we added a English translation of the different food diaries.

Round 2

Reviewer 1 Report

The authors have amended the manuscript and addressed comments and included methods about the oral food challenge, rectified the issue regarding consent and also extended the discussion, so I am now recommending it is accepted for publication. 

Reviewer 2 Report

There are only some minor linguistic mistakes (it's --> it is).

All other answers were satisfactory